# Understanding the use of patient-reported data by health care insurers: A scoping review

**Anne Neubert**[1,2], **Óscar Brito Fernandes**[3,4]*, **Armin Lucevic**[3,4], **Milena Pavlova**[5], **László Gulácsi**[3], **Petra Baji**[3], **Niek Klazinga**[4‡], **Dionne Kringos**[4‡]

1 Department of Orthopaedics and Traumatology, Medical Faculty, Heinrich Heine University Düsseldorf, Düsseldorf, Germany, 2 Institute for Health Service Research and Health Economics, Centre for Health and Society, Heinrich-Heine-University, Düsseldorf, Germany, 3 Department of Health Economics, Corvinus University of Budapest, Budapest, Hungary, 4 Department of Public and Occupational Health, Amsterdam UMC, University of Amsterdam, Amsterdam Public Health research institute, Amsterdam, The Netherlands, 5 Department of Health Services Research, CAPHRI, Maastricht University Medical Center, Faculty of Health, Medicine and Life Sciences, Maastricht University, The Netherlands

☯ These authors contributed equally to this work.
‡ These authors also contributed equally to this work.
* obritofernandes@uni-corvinus.hu

**Data Availability Statement:** All relevant data are within the paper and its Supporting Information files.

## Abstract

### Background

Patient-reported data are widely used for many purposes by different actors within a health system. However, little is known about the use of such data by health insurers. Our study aims to map the evidence on the use of patient-reported data by health insurers; to explore how collected patient-reported data are utilized; and to elucidate the motives of why patient-reported data are collected by health insurers.

### Methods

The study design is that of a scoping review. In total, 11 databases were searched on. Relevant grey literature was identified through online searches, reference mining and recommendations from experts. Forty-two documents were included. We synthesized the evidence on the uses of patient-reported data by insurers following a structure-process-outcome approach; we also mapped the use and function of those data by a health insurer.

### Results

Health insurers use patient-reported data for assurance and improvement of quality of care and value-based health care. The patient-reported data most often collected are those of outcomes, experiences and satisfaction measures; structure indicators are used to a lesser extent and often combined with process indicators. These data are mainly used for the purposes of procurement and purchasing of services, quality assurance, improvement and reporting, and strengthening the involvement of insured people.

### Conclusions

The breadth to which insurers use patient-reported data in their business models varies greatly. Some hindering factors to the uptake of such data are the varying and overlapping

**Funding:** The participation of AL, DK, LG, NK, OBF and PB occurred within a Marie Skłodowska-Curie Innovative Training Network (HealthPros – Healthcare Performance Intelligence Professionals) that has received funding from the European Union's Horizon 2020 research and innovation programme under grant agreement Nr. 765141 (https://healthpros-h2020.eu). The funder had no role in study design, data collection and analysis, decision to publish, or preparation of the manuscript.

**Competing interests:** The authors have declared that no competing interests exist.

terminology in use in the field and the limited involvement of insured people in a health insurer's business. Health insurers are advised to be more explicit in regard to the role they want to play within the health system and society at large, and accommodate implications for the use of patient-reported data accordingly.

## Introduction

In recent years, there has been an increased focus among policy makers, health insurers, and care providers on maximizing value and reducing waste in healthcare. In this regard, two central concepts have emerged: quality of care (QoC) and value-based healthcare (VBHC). QoC emphasizes the importance of care delivery that is compliant with the best possible standards, taking into consideration the cultures in a society, and aligned with the healthcare service users' needs, expectations, and preferences [1–3]. Nowadays, it is commonplace to associate VBHC with care quality. Although a key component of quality, it is not necessarily the mainstream culture for measuring thereof. The VBHC agenda, similarly to the QoC, puts forward patients' values regarding health and care outcomes, stressing their involvement in decision-making processes [4]. The construct of patient-centeredness emerges as a sub-dimension of those two concepts (QoC and VBHC) [5]. However, the inclusion of a people-centered perspective in VBHC is not without tensions as VBHC is a concept derived from management theories, with a clear conceptual focus on costs [6]. Hence, there can be a tension between the business model of a health care insurer oriented to optimizing the value for individual patients/insured versus optimizing the health of a population such as the group of individuals that pay their premium for the insurance. To strengthen people-centeredness and strive towards QoC and VBHC, health system stakeholders (e.g. health care insurers and care providers) should commit to the value agenda supported by intelligence on the healthcare system users' needs, expectations, and preferences [7–9]. Hence, patient-reported data have become crucial to gain insight on one's voice and inform the decisions of those key stakeholders.

The most commonly collected patient-reported data are those related to outcomes and experiences of care. Patient-reported outcome measures (PROMs) can be either used to measure the outcome of a specific disease or to assess the general health status of a person, and they are commonly used by clinicians and hospitals [10]. Other uses are those related to drug reimbursement schemes [4,11,12] and health technology assessment [13]. On the other hand, patient-reported experience measures (PREMs) refer to a person's experiences while interacting with the healthcare system (e.g. to receive care) [9].Research and policy discussions on PROMs and PREMs have predominantly focused on the use of patient-reported data by healthcare providers to improve clinical practice [14–16]. For example, the work of the International Consortium for Health Outcomes Measurement (ICHOM) has contributed to setting international standards for outcome measures that matter most to patients on varying diseases [17]. In parallel, the Organisation for Economic Co-operation and Development (OECD) is promoting the PaRIS project [18,19], which focuses on indicator surveys that capture PROMs and PREMs of people with breast cancer, hip- and knee surgery, or mental health problems, as well as the development of new tools to people with multiple chronic conditions treated in primary care settings. However, less is known about the use of patient-reported data by health insurers in supporting people-centeredness for QoC and VBHC [20,21]. An investigation of this issue is opportune given the evolving role of insurers across health systems. Health insurers are no longer solely focused on cost containment and cost-effectiveness, but also on

adequate health service design and planning of improved health of the (insured) population [22]. Hence, research on the use of patient-reported data by health insurers can help to determine to what extent health insurers respond to the insurees' needs and preferences [8,20].

Our study aims to: 1) map the evidence on the use of patient-reported data by health insurers; 2) explore how patient-reported data are utilized; and 3) elucidate the motives of why patient-reported data are collected by health insurers.

## Methods

We conducted a scoping review design following the Preferred Reporting Items for Systematic Reviews and Meta-Analysis extension for scoping reviews (PRISMA–ScR) [23] (S1 File). The procedures to conduct the scoping review were disseminated across the research team for feedback and improvement. To enhance the quality of the methodology used, we based the review on the stepwise methodological framework suggested by Arksey and O'Malley (2005) [24], while also taking into account the recommendations of other authors [25,26].

### Search for relevant studies

We performed a two-tier search: systematic and non-systematic. The search criteria were discussed among the researchers before the start and during the search period, as suggested by Levac et al. (2010) [26]. The systematic search was conducted between May 21st and May 26th, 2019. In total, 11 databases were searched: PubMed, Embase, Health Systems Evidence, NICE, JSTOR, Emerald, Wiley, Cochrane Library, PDQ-Evidence, NIHR Journals Library, and EBSCO/Health Business Elite. The following search terms were used as subject headings or free-text words, including synonyms and closely related words: ("health insurer," or "health insurance," or "private health plans," or "medical care insurance,") and ("patient-reported data," or "consumer reported data," or "PREMS," or "PROMS," or "consumer satisfaction," or consumer preferences," or "consumer feedback"). The choice for the search terms was based on an initial quick literature scan and discussion among the researchers. During the initial exploratory searches, we have observed that the terminology on patient-reported data varied widely. Thus, we informed our search strategy with key terms used in systematic reviews on patient-reported measures (e.g. [27–29]). We limited the search to keywords found in title and abstract to minimize the number of off-topic hits, which otherwise would have been unmanageable. The search strategy was adapted to each database and can be found as supplemental information (S2 File). We included all peer-reviewed study types that were written in the English, German, or Dutch language. The search was not time bounded except for the JSTOR database. Not limiting the timeframe was producing a large number of hits off-topic to this review, which revealed to be unmanageable; hence, we limited the search from the year 2000 onward, where documents of potential relevance to the screening process started to emerge.

For the non-systematic search, we included relevant grey literature such as webpages of insurers and third-party reports (S3 File). The search was performed between May 8th and June 18th, 2019. The documents retrieved were identified through online searches, reference mining, and recommendations from experts. The latter refers to contacts we have established via email (e.g. health insurers, insurance associations/federations, consultancy firms, or patient advocates) to direct us towards potential documents to complement the internet search. In total, 23 emails were sent to institutions that we believed could bring clarity or provide further information on top of what we had read on their webpages; 9 answers were received until the 18th of August 2019. A reminder was sent to all unanswered emails; we have received no reply to 14 emails and closed further contacts by the end of August 2019.

## Selection of studies

After removing duplicates, the selection of studies was independently and blindly performed by Anne Neubert (AN), Óscar Brito Fernandes (OBF), and Armin Lucevic (AL) with the open-source application Rayyan [30]. Prior to the screening, the reviewers grounded the eligibility criteria based on two principles: 1) to exclude the studies that did not highlight patient data (e.g. PREMs or PROMs) and its use by health insurers or information on how insurers respond to patients' expectations, needs, and preferences; and 2) to exclude the studies where the setting was not a high-income country, based on the assumption that a health insurance system in developing countries might differ greatly from that of developed countries (e.g. on the extent to which patient-reported data are employed), and thus limiting the generalizability of the study's findings. The reviewers also agreed that if two researchers agreed on inclusion (exclusion) of a document, the document would be included (excluded) for full text reading. In cases where all three researchers had divergent opinions, the researcher who classified the document with *maybe* (a possibility with Rayyan) had to make a blinded final decision for inclusion or exclusion without prior information on arguments that supported the decision of the other two researchers involved. If needed, discussions with non-scoring researchers were allowed. All three researchers first screened the publications by title and abstract/executive summary. The full text review that followed was performed by AN and OBF; AL was involved as tiebreaker, if any.

## Data charting and analysis

All documents retained for analysis were subject to content extraction into a data charting form and synthesis of the following information: author(s) and year of publication, study setting (country), a brief description of the content, the indicator(s) highlighted in the study, and the use and function of those indicators by health insurers. We organized the list of indicators following the Donabedian's healthcare quality model (structure-process-outcomes) [1,31] given how widespread and familiar this model is across health systems and its stakeholders. This option could also facilitate a first approach to organize scattered information about the purposes and uses of patient-reported data by health insurers. Data mapped under *structure* highlight measures regarding the context and setting wherein care is delivered, and data under *process* highlight the interactions between a person and providers throughout the care trajectory; regarding data under *outcomes*, we organized information as clinical measures (referring to the diagnosis, treatment, and monitoring of a person), and patient-reported measures (PROMs, PREMs, and satisfaction measures). The focus of our work is on patient-reported data, but by using structure, process, and clinical measures as ancillary indicators in our work, we expected to have a better understanding on how patient-reported measures are used (as standalone or combined with other measures). The charting form was agreed among the research team; both AN and AL independently identified the relevant information in the studies to populate the table (S4 File).

## Study validity and reliability

To improve the validity of the review we considered two types of triangulation: tier triangulation (related to the researchers) and data triangulation. To support the former, the research team maintained an open discussion and iterative approach across all phases of the study. In addition, our review triangulated data accessed from different sources [32,33] and all searches and data analysis were thoroughly documented [34]. Given that the data collected differed greatly in breadth and depth, we followed the suggestion of Silverman (2009) [25] of synthesizing evidence with the support of a table to enhance the reliability of the review.

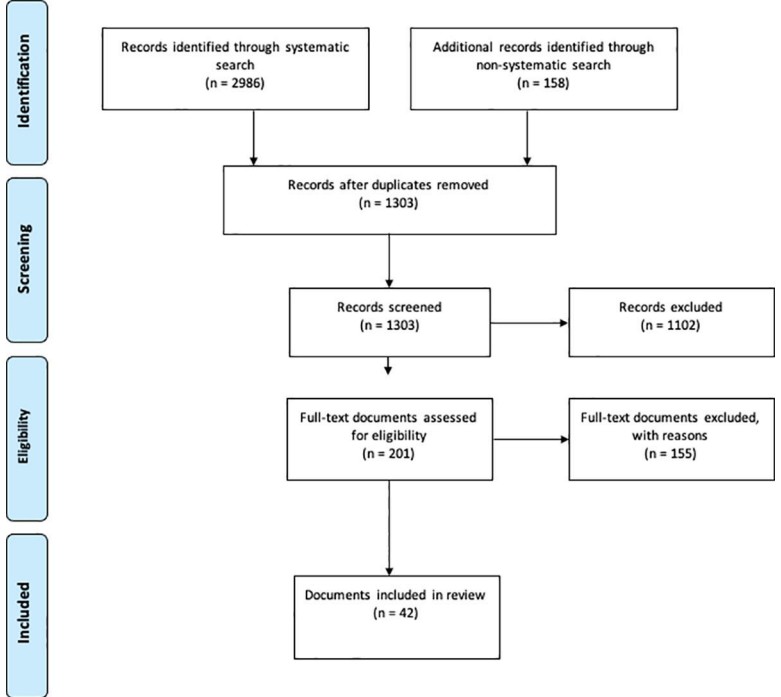

**Fig 1. PRISMA chart of document selection.**

## Results

The systematic search initially generated 2986 articles and the non-systematic search 158 documents, including grey literature and email correspondence (Fig 1). After the screening process, 42 documents were considered eligible for inclusion: 15 retrieved from the systematic search and 27 from the non-systematic search. From the latter group, 17 documents were classified as grey literature.

### Characteristics of the documents

The documents included in the study covered the period from 1996 to mid-2019 (Table 1). The majority were written in the English language (n = 30), followed by German (n = 11), and Dutch (n = 1). More than a quarter of the documents portrayed the situation in Germany [35–46], followed by the USA [47–57], and the Netherlands [58–68]. Six documents discussed a multiple-country setting [69–74], one highlighted the UK context [75], and one document had no specific country attached [76]. Among the journal articles, 13 documents were quantitative studies and six were qualitative; also, seven articles were classified as non-empirical, such as reports, commentaries, and summaries.

### What kind of data do insurers use?

The use of PROMs was the most spread across the documents [27,35,37,38,42–44,47–52,54–57,59,62,63,65,67,68,72–76] relative to PREMs [35,36,38,39,41,44,46,50,52,57,59,60,71] or satisfaction measures [35,37,38,41,44,50,58,61,66]. Generic measures on patient satisfaction were often complemented with specific PREMs or PROMs [35,37,38,41,44,50]. Often, PROMs were employed in combination with clinical indicators or in combination with patient-reported outcome-based performance measures [38,49,52,54,62,67,73,76]. The use of structure

**Table 1. Characteristics of the documents included in the study (N = 42).**

| Characteristic | N | % |
|---|---|---|
| **Year of publication** | | |
| Prior 2005 | 1 | 2% |
| 2005–2009 | 7 | 17% |
| 2010–2014 | 17 | 40% |
| 2015–2019 | 17 | 40% |
| **Language** | | |
| English | 30 | 71% |
| German | 11 | 26% |
| Dutch | 1 | 2% |
| **Setting** | | |
| Germany | 12 | 29% |
| The Netherlands | 11 | 26% |
| USA | 11 | 26% |
| United Kingdom | 1 | 2% |
| Multiple | 6 | 14% |
| Non-specific | 1 | 2% |
| **Type of indicator** | | |
| Structure | 6 | 14% |
| Process | 10 | 24% |
| Outcome | 37 | 88% |
| Clinical | 14 | 33% |
| Patient-reported outcome measure | 27 | 64% |
| Patient-reported experience measure | 12 | 29% |
| Patient satisfaction measures | 9 | 21% |
| Non-specific | 3 | 7% |
| **Use and function of indicators** | | |
| Procurement and purchasing of healthcare services | 17 | 40% |
| Quality reporting | 11 | 26% |
| Involvement of insured people | 8 | 19% |
| Performance assessment of providers | 6 | 14% |
| Profiling | 6 | 14% |
| Quality assurance and improvement | 4 | 10% |
| Product/Program development | 3 | 7% |

The percentages in *year of publication*, *language*, and *setting* have been rounded and may not total to 100% (rounding error).

indicators by health insurers, such as the availability of specific disease programs or the existence of quality assurance certification, were less frequent [38,42,52,53,67,70] and often used in combination with process indicators [38,42,52,53,67]. On the other hand, process indicators [35,37,38,42,52–54,66,67,71] and clinical outcome indicators [38,41,42,49,52,54,58,62,64,66,67,71,73,76] were frequently mentioned.

## How do insurers use patient-reported data?

Based on the uses and fuctions of patient-reported data among selected documents, we identified 17 documents (40%) discussing the procurement and purchasing of healthcare services [42,43,51–53,56,57,60,64–69,71,74,76]. Quality reporting was highlighted in 11 documents

(26%) [35–37,39,41,44,46,50,60,70,76], and four more (10%) focused on quality assurance and improvement [42,63,72,75]. Other key uses and functions were those of strengthening the involvement of insured people [38,55,60,63,72–75], measuring the performance of providers [38,41,59,60,71,76], profiling [40,47,48,54,58,62], and the development of products/programs [45,49,61].

**Procurement and purchasing of healthcare.** The use of PREMs in the context of procurement and purchasing of healthcare services was available in Delnoij et al. (2010) [60], Cashin et al. (2014) [71], and Damberg et al. (2014) [52], whereas the use of satisfaction measures was discussed in Dohmen and van Raaij (2019) [66]. The use of PROMs was most frequently discussed [42,43,51,52,56,57,65,67,68,74,76], and only Damberg et al. (2014) [52], Klakow-Franck (2014) [42], and Moes et al. (2019) [67] discussed the broader use of structure, process, and outcome indicators in procurement and purchasing processes.

Selective contracting was discussed in five documents (12%) [60,67–69,74]. In general, selective contracting refers to the contractual agreement between a health insurer and a provider, where the former selects those providers that meet certain QoC expectations. The inclusion of QoC indicators is highly dependent on the availability of data; hence, the most common data used in these contracts are based on volume and costs [66,69], and only recently some incorporate PROMs (and in a lesser extent, PREMs) [68,74]. The use of structure, process, and outcome indicators for the purpose of selective contracting was discussed in Moes et al. (2019) [67]. A pitfall, however, relates to the varying terminology used for selective contracting. The term 'selective contracting' was mainly deployed in the Dutch literature [60,67–69,74]. Other terms were 'outcome-based purchasing' [64], 'quality contracting' (predominant in German literature) [42,43], and 'value-based purchasing' or 'payments' (predominant within the US literature) [51–53,56,57].

One of the main objectives for selective contracting was that of value-based purchasing or value-based payment programs (VBP). Notwithstanding, improvements on QoC at large were also an objective, with a special focus on dimensions such as effectiveness, efficiency, and safety. Conversely, patient-centeredness was not one of the major areas to strive for and it was commonly discussed as appropriateness of care (e.g. reduction of overuse and underuse of care) [68,69]. In general, different patient-reported data were required for selective contracting [60,67–69,74] and pay-for-performance programs (P4P) [56,71]. The former required data that enabled comparisons across providers to contract those that are performing best; the latter required data that enabled health insurers to compare the performance of a provider with a predetermined target, norm, or past performance [60].

**Quality assurance, improvement, and reporting.** The focus on quality assurance, improvement, and reporting was frequent among the documents, with higher frequency for quality reporting of the performance of providers (of mainly inpatient services, but of late also outpatient services). Two perspectives on the reporting of the QoC of providers emerged: 1) as an ancillary instrument to inform the decisions of insurees; and 2) as a means of supporting and enhancing quality improvement via the benchmark of providers [70,76]. Terminology about quality reporting was also varying across the documents: 'public reporting' [70], 'hospital ranking' [36], 'quality reports' [35], 'doctor assessment portals' [41,46], or 'performance comparison' [76]. Different tools were discussed for quality reporting, such as the Dutch Consumer Quality Index [60] and the German Patients' Experience Questionnaire [41]. PREMs were often used for quality reporting [35,36,39,41,44,46,50,60], as were PROMs [35,37,44,50,76]. The use of structure and process measures for quality reporting was featured far less, relative to PROMs and PREMs [35,37,70].

**Prediction models.** Our findings suggested that health insurers use prediction modelling to forecast and profile enrollees who are likely to incur high medical costs [40,47,48,54,58,62].

The documents often applied the term 'self-reported data' when referring to health behavior, healthcare utilization, morbidity, and health status data, which were often combined with claims data. For example, Fleishman et al. (2006) [48] and Hornbrook and Goodman (1996) [47] used PROMs in their profiling studies, namely the RAND-36 and SF-12.

**Other purposes.** Alongside the uses of patient-reported data by health insurers reported so far, other uses were identified, such as the involvement of insured people in decision-making (n = 8; 19%) and the development of products/programs (n = 3; 7%). The first stresses the role of a health insurer in research by granting access to data (e.g. claims data) [72] and the development of novel PREMs and PROMs that are both fit for purpose and use [55,63,75]; the second relates to the role of a insurer in the development or co-creation of healthcare projects that incorporate the use of patient-reported data, such as those portrayed by Nickel et al. (2010) [38] and Franklin et al. (2017) [55].

## Why do insurers use patient-reported data?

**Quality of care.** The focus of most of the uses of patient-reported data was that of QoC at large or that of a particular dimension of QoC, such as effectiveness, efficiency, access, patient-centeredness, equity, or safety [2,3]. Effectiveness of care was relevant mostly in relation to cost-effectiveness [38,45,47,50,53,55] and effectiveness of care [49,51,54,58,66,70,76]. Efficiency was a dominant topic with a focus on economic efficiency, cost-efficiency, allocative, and technical efficiency [38,43,52,56,69,76]. Some sources used efficiency in relation to the efficient targeting of patients with high healthcare needs [54].

Patient-centeredness was often mentioned in relation to the appropriateness of an intervention or service [52,68,69], interventions that are centered around the patient [49], and as a goal of using PROMs [56]. In addition, patient safety was discussed, alongside to effectiveness, as key to selective contracting and measuring the quality of a provider. Other authors employed the term 'patient safety' to judge the performance and quality of health services for diverse purposes [42,57,67], or to refer to requirements of treatments to guard patients' safety [64].

Access was often a topic in relation to equity [40] and accessibility of healthcare for people with disabilities [43]. Equity was the least mentioned in the documents retrieved.

**Value-based healthcare.** VBHC was mentioned as an important reason for employing patient-reported data. One approach viewed VBHC as the value of a service for a patient, whereas a second approach focused on the purchasing or payment methods. Value-based payments [54–57,66] and value-based purchasing [52,53] were described in some of the documents. For example, Dohmen and van Raaij (2019) [66] showed how Zilveren Kruis, a Dutch health insurer, was piloting a method (best-value procurement) to purchase services from providers that do not only focus on volume and cost. Similarly, Squitieri et al. (2017) [56] explored how to integrate PROMs in value-based payment reforms to measure the performance of service providers from a patient's perspective.

## Discussion

In this study, we looked at the what, how, and why of health insurers using patient-reported data. Our findings inform that the patient-reported data most often collected by health insurers are those of PROMs, followed by PREMs and satisfaction measures. These data are mainly used for the procurement and purchasing of services; quality assurance, improvement, and reporting; and strengthening the involvement of insured people. Health insurers use patient-reported data for assurance and improvement of QoC and VBHC.

The findings of our study suggest that the use of patient-reported data by health insurers is common and centered on PROMs, often combined with clinical outcomes or process

measures. PREMs data, albeit used to a lessen extent, were somewhat depicted in the documents analysed. These data are central to support health insurers towards the procurement and purchasing of services (including the practice of selective contracting), and quality assurance, improvement, and reporting with the purpose of supporting QoC improvement and VBHC. However, the breadth to which insurers use such data in their business models varies greatly. Some factors may hinder the use of patient-reported data on a larger scale. First, requiring from the insurer side data collection in a timely fashion, including patient-reported data, entails the ability of an insurer to invest in a robust health information system which could reduce fragmentation of data flow between the insurer and care providers [77]. Second, the culture of an insurer, as well as the organization's corporate values, may influence the role of the insurer in a healthcare system (and in the society at large) and the perception of the usefulness of patient-reported data as key to inform business practices and decision-making [78,79]. Third, contextual factors, such as country-specific legislation, data protection regulation, the organization of the healthcare system, and market competition may influence the diffusion of the use of patient-reported data across a health insurer's business. As suggested by Klose et al. (2016) [80] and Brito Fernandes et al. (2020) [8], the knowledge about patients' needs, preferences, and experiences could help organizations such as health insurers in developing and optimizing a patient-centered approach.

Selective contracting and P4P programs that use patient-reported data such as PROMs and PREMs are still under-developed, albeit some initiatives. For example, insurers in the Netherlands are being encouraged by governmental regulation to assume a role of active purchasers [81]. If health insurers are enhancing their procurement and purchasing practices in relation to QoC, health systems could evolve from demand-driven to quality-driven purchasing, as well as from performance-based towards quality-rewarding payments. This would entail that purchasers change from passive funders of care to an active promoter of QoC, who base the financing of healthcare services on good quality and on what is of value to (insured) people and communities [60,73,74].

In relation to quality assurance, improvement, and reporting, we found that health insurers have a growing role in driving the performances of care providers. This entails giving insurees the possibility of choosing providers based on information related to quality. This may influence the decision-making of an insured person when selecting a care provider, and to a limited extent influence the QoC provided [81]. However, patients often do not rely on quality reporting to support their decision-making, partly because they perceive these initiatives not driven by quality concerns but rather by political interests [82]. Hence, health insurers should further commit to involve insurees in initiatives that develop and report on measures that resonate with insurees. Further, health insurers should not only concentrate on reporting the quality of providers, but also align incentives that support the investigation of root causes of poor quality at a provider-level [82].

Our findings highlight a large heterogeneity of the terminology used in the literature. This was also identified by Desomer et al. (2018) [83]. The extent to which it may have hindered a clearer picture of how health insurers use patient-reported data remains unanswered. This wide variation in the conceptualization of PROMs and PREMs could suggest that these measures are not yet optimized to fully address a wide scope of need for information across actors [84]. In addition, methodological challenges (e.g. fit for risk-adjustment or a people-centered approach to developing such measures) offer another layer of complexity to the conceptualizations of such data. This context also sets the opportunity for new measures to arise, such as that of patient-reported outcome-based performance measures [51,56] and preference-based PREMs [8].

## Strengths and limitations

The main strength of our study is its design, which enabled us to find literature that is highly scattered and unstructured. The findings of our review should be interpreted in light of some limitations. The heterogeneity of terminology, the use of an unsystematic search component, and language restriction may have introduced bias. To mitigate possible effects, the search strategy was informed by (but not limited to) the terminology used in other systematic reviews related to patient-reported data; we also assessed the extent to which documents retrieved via unsystematic search were aligned with those retrieved via systematic search. Also, limiting our search to high-income countries may have also introduced general bias. On the one hand, we did not consider studies from low-middle income countries because the use of patient-reported data in such contexts is yet limited [85]; on the other hand, we acknowledge that even in high-income countries, the extent to which patient-reported data are used may vary greatly, considering the role and involvement of a health insurer in the health system. Finally, given the study design, generalizability of results is limited; for example, contextual factors (e.g. the organization and digitalization of the healthcare system) vary greatly, and the extent to which these affect the uses and applications of patient-reported data are unknown.

## Conclusions

The breadth to which insurers use patient-reported data in their business models varies greatly across countries. Health insurers are actively using patient-reported data to enhance QoC and VBHC, predominantly through procurement and purchasing of healthcare; quality assurance, improvement and reporting; and the involvement of insured people. However, our study highlights three key aspects that hinder a more robust use of such data in a health insurer's business. First, the insurers' use of patient-reported data is affected by a large technological and methodological heterogeneity that inhibits the transferability of innovative and effective initiatives across contexts. Second, the varying terminology of constructs used by the many stakeholders with whom an insurer interacts. Third, the involvement of insured people by insurers in the development of patient-reported measures and decision-making in regard to a health insurer's strategy and practices is still limited. To overcome these hindering factors, health insurers are advised to be more explicit in regard to the role they want to play within the health system and society at large. In addition, health insurers should have a clear scope about the use and actionability of patient-reported measures, and further involve insurees to the extent where it is feasible and deemed necessary. For many years now, there is a generalized consensus among healthcare providers and professionals for a greater involvement and engagement of people in decision-making towards a more people-centered health system. Albeit significant advances, we still fall short on that cornerstone. The extent to which lessons learned by health systems could be used and known obstacles could be overcome by health insurers remain overlooked and deserve further research.

## Supporting information

**S1 File. Preferred reporting items for systematic reviews and meta-analyses extension for scoping reviews (PRISMA-ScR) checklist.**
(DOCX)

**S2 File. Full search strategy for each database.**
(PDF)

**S3 File. Overview of the organizations whose websites were consulted during the non-systematic search.**
(PDF)

**S4 File. Charting form of the documents retrieved in the systematic and non-systematic search following a structure, process, outcome organization.**
(XLSX)

## Acknowledgments

An initial version of this research was presented in the master's dissertation of AN at Maastricht University.

## Author Contributions

**Conceptualization:** Anne Neubert, Óscar Brito Fernandes, Armin Lucevic, Milena Pavlova, László Gulácsi, Petra Baji, Niek Klazinga, Dionne Kringos.

**Data curation:** Anne Neubert.

**Formal analysis:** Anne Neubert, Óscar Brito Fernandes.

**Funding acquisition:** Niek Klazinga, Dionne Kringos.

**Investigation:** Anne Neubert, Óscar Brito Fernandes, Armin Lucevic.

**Methodology:** Anne Neubert, Óscar Brito Fernandes, Armin Lucevic, Milena Pavlova, László Gulácsi, Petra Baji, Niek Klazinga, Dionne Kringos.

**Supervision:** Milena Pavlova, Petra Baji, Niek Klazinga, Dionne Kringos.

**Validation:** Anne Neubert, Óscar Brito Fernandes, Armin Lucevic, Milena Pavlova, Petra Baji.

**Visualization:** Anne Neubert, Óscar Brito Fernandes.

**Writing – original draft:** Anne Neubert, Óscar Brito Fernandes.

**Writing – review & editing:** Anne Neubert, Óscar Brito Fernandes, Armin Lucevic, Milena Pavlova, László Gulácsi, Petra Baji, Niek Klazinga, Dionne Kringos.

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
