## [Decision Letter · Decision Letter 0]

7 Sep 2020

PONE-D-20-09776

Why, what and how do health insurers use patient-reported data? Results of a scoping review

PLOS ONE

Dear Dr. Brito Fernandes,

Thank you for submitting your manuscript to PLOS ONE. After careful consideration, we feel that it has merit but does not fully meet PLOS ONE’s publication criteria as it currently stands. Therefore, we invite you to submit a revised version of the manuscript that addresses the points raised during the review process.

The main issue is that the frameworks used should be described better and, especially, applied more thoroughly throughout the manuscript. Also address inconsistencies in the scoping review methodologies used. Furthermore, focus on a critical discussion of the VBHC concept, address differences among health insurance systems in high-income countries, provide definitions or at least descriptions according to the reviewers’ comments (e.g. on clinical quality, selective contracting), consider revising the title (I am also not a native English speaker but it feels a bit odd, should “what” perhaps be “which”?) Please do not describe or discuss new literature/evidence from other studies in the results section but either in the methods or discussion (or introduction if suitable). Also consider to include only a selection of the main table in the main manuscript and include the remainder/full table in an appendix. Finally I am not entirely sure on the aspects included in the results section for “how”, as e.g. selective contracting is more of a purpose instead of exactly describing how patient-reported data are used, and ensure that the order of the different topics of what, how and why are always the same throughout the manuscript (e.g. in the title, how they are mentioned in the introduction, subsections in results and discussion).

We look forward to receiving your revised manuscript.

Kind regards,

Mathieu F. Janssen, Ph.D.

Academic Editor

PLOS ONE

Journal Requirements:

Additional Editor Comments (if provided):

The main issue is that the frameworks used should be described better and, especially, applied more thoroughly throughout the manuscript. Also address inconsistencies in the scoping review methodologies used. Furthermore, focus on a critical discussion of the VBHC concept, address differences among health insurance systems in high-income countries, provide definitions or at least descriptions according to the reviewers’ comments (e.g. on clinical quality, selective contracting), consider revising the title (I am also not a native English speaker but it feels a bit odd, should “what” perhaps be “which”?) Please do not describe or discuss new literature/evidence from other studies in the results section but either in the methods or discussion (or introduction if suitable). Also consider to include only a selection of the main table in the main manuscript and include the remainder/full table in an appendix. Finally I am not entirely sure on the aspects included in the results section for “how”, as e.g. selective contracting is more of a purpose instead of exactly describing how patient-reported data are used, and ensure that the order of the different topics of what, how and why are always the same throughout the manuscript (e.g. in the title, how they are mentioned in the introduction, subsections in results and discussion).

Reviewers' comments:

Reviewer's Responses to Questions

**Comments to the Author**

1. Is the manuscript technically sound, and do the data support the conclusions?

Reviewer #1: Yes

Reviewer #2: Yes

Reviewer #3: Partly

2. Has the statistical analysis been performed appropriately and rigorously? 

Reviewer #1: N/A

Reviewer #2: N/A

Reviewer #3: N/A

3. Have the authors made all data underlying the findings in their manuscript fully available?

Reviewer #1: Yes

Reviewer #2: Yes

Reviewer #3: No

4. Is the manuscript presented in an intelligible fashion and written in standard English?

Reviewer #1: Yes

Reviewer #2: Yes

Reviewer #3: Yes

5. Review Comments to the Author

Reviewer #1: The authors conducted a scoping review in order to map the evidence how insurers use patient-reported data. In doing so, the investigators highlighted patient-centeredness with respect to quality of care and value-based healthcare. The investigators specifically sought to understand why and how these data are collected and utilized. The review was conducted using the PRISMA extension for scoping reviews, and the analysis was framed using a value chain concept.

The scoping review was conducted in a rigorous and transparent manner. The manuscript had a logical flow and the investigators explained, and provided citations for, the basis/framework for each step of the process. The manuscript was well-written, too. As a result, I do not perceive any areas of weakness with regard to the design or implementation of the manuscript.

Although this may be outside of the scope of the manuscript, I was most interested in the potential “next steps” of this work based on the manuscript’s limitations. One area that might merit further thought is with regard to how stakeholders could implement a more consistent terminology of constructs. Furthermore, did the investigators think about broad differences in the way that citizens residing in different countries might have conceptualized the role of patient-reported measures and decision-making based on their nation’s health system and prevailing ideology regarding health? Finally, what role within the health system and in society should a health insurer play and is that role different in different nations? While these are questions have no easy answer, the investigative team is well-equipped to provide their opinions.

Reviewer #2: The manuscript entitled “Why, what and how do health insurers use patient-reported data? Results of a scoping review” is well written and highlights an important topic of current interest, so far not studied to any extent. Given the breadth and variety of the topic, a scoping reviews seems like the best choice of method. The review process is well described and seems to be conducted in a sound way.

The title askes “why, what and how”. I understand why and how, and I guess “what do health insurers use patient-reported data” means which patient-reported data is being used, though I am not proficient enough in English to know if that is a grammatically correct expression? If “what” refers to something else, maybe it could be further explained?

Though QoC is by now a concept widely used globally, and the use of indicators of structure, process and outcome commonplace in evaluations, VBHC is more of a management trend (in some countries more than others), following earlier trends like TQM and lean production, and as such also subjected to criticism not acknowledged in the manuscript.

The decision to exclude the studies where the setting was not a high-income country, based on the assumption that a health insurance system in developing countries might differ greatly from that of developed countries, seems fair. However, also among high-income countries health insurance systems and their involvement in healthcare systems differ greatly, not acknowledged by the authors. Though I guess out of the scope of the present study it would be interesting to study the difference in the use of patient-reported outcomes divided by differences in health insurance systems and their involvement in healthcare systems.

In the section “Data charting and analysis” it is described that indicators are organized in a traditional structure-process-outcome approach. For those not familiar with QoC and Donabedian, this might need an explanation, or at least a reference, maybe [1]?

In Table 1, I do not seem to find an explanation for the star* in PREM*? Furthermore, the definition of clinical quality is not stated. I am well aware of the problems with categorizing in this field, and reading Table 1 makes we wonder if all patient-reported data in the column PROM is actually reported as outcomes? I suspect some to be reported more as “patient characteristics”. “Health status” is a very generic term. I guess that the terms included in the table are the actual terms used in the papers (could be described), but if it is the authors interpretations, they should be defined/explained in more detail. Moreover, I wonder if only PROMS, PREMS and patient satisfaction are patient-reported? I guess that though the title implies that only patient-reported data are included in the study, the data regarding structure and process are mainly not patient-reported, but rather supportive data for your study. However, this is not very clearly stated.

Some specifics: In row 1, it is stated “RAND-36 survey, which included information on…”. I suspect that is not true, RAND 36 is the original, free version of SF-36, and does not include self-reported morbidity etc.

The concept of selective contracting needs a short description, I think many readers are not familiar with the concept.

I find your Discussion and Conclusion very apprehensive. The recommendation “In addition, health insurers should have a clear scope of the use of patient-reported measures” made me reflect on the fact that for so many years healthcare professionals have failed to accomplish just that. Could health insures achieve what health professionals could not, or will it be even more difficult for health insurers…

Reviewer #3: General comment:

In general, the topic of the article is well picked and scientifically relevant. The use of a scoping review is suitable for the aim of the article. Research question was addressed in methods and result section.

However, there are some inconsistency in methods and results, so that question regarding external validity and pictured landscape of health insurer's use of patient reported data arises. In my opinion the concerns could be solved by clarification/specification of the explanations given. The main critic relies on the 13 pages table within the main text. This should be solved and adjusted due to the understanding and focus of the article. Also the frameworks used (structure-process-outcome as well as value chain) should not only be described better, but also be more applied within the text.

In detail feedback:

Methods:

Scoping review methodology: The article stated that it was following Arksey and O’Malley (2005) stepwise framework. In the next section, they referred to the framework of Levac et al (2010) and later on to Silverman (2009), which seems a little bit inconsistent. Levac et al (2010) recommended and suggested for instance a update of the Arksey and O’Malley framework. The question arises, why different methodology were used or not only Levac was the entire foundation of the methodology. Probably it should be enough to frame this differently saying that Arksey was the basis, but recommendation of other authors was taken into account as well.

Structure-process-outcome approach: there was no explanation included within the methods section about the approach (even though it was claimed as ‘traditional’ and as the indicators for the research question). There was also no explanation of the rationale behind, the application in data extraction (e.g. why are outcomes differentiated into four subtypes and others not and are the suitable) and the suitability for the research question itself. However, it was referred to it in the result section entirely (good!), but, therefore, an introduction/explanation before is needed.

The search strategy for NIHR Journals Library, PDQ-Evidence, EBSCO/Health Business Elite and Cochrane Library is not attached within the supplement file. This should be added due to the PLOS One requirements. Moreover, NICE is stated within the supplement file, but not listed in the text for the systematic search.

Row 132-135: Rationale for expert interviews with Hungarian health insurance is unclear. Is this adoptable to other settings? Why was then the Hungarian language not included? Is asking only one insurance company not quite biased?

Time limit for JSTOR: why was a time limit for JSTOR applied, but not for the other databases? Argumentation is flawed here.

Focus on PROMs and PREMs: in the introduction it is stated that a specific focus is on PROMs and PREMS. This was reflected by using the terms specifically in the search strategy for Pubmed and Embase. However, the focus was mainly seen in the introduction, later on the results showed a wider perspective. I would recommend to reframe it. Otherwise questions about possible bias, rational and differentiation towards other methods arises.

Search terms: Search terms were mainly limited to Title/Abstract. The authors argued in the limitation section that Mesh terms/indexed words were not available for their specific topic. The question arises, why then the limitation on Title/Abstract was applied to mitigate the obstacles (especially because it is a scoping review). Furthermore, the limitation of Mesh Terms is in general obvious due to the time gap between new topics and the indexing process, so research in general should not entirely rely on them.

Results:

The presented table is covering 13 pages in total in the main text. In my opinion, this is way too long for a scientific journal article and not benefitting the understanding of the article. I would highly recommend to move the table towards the supplemented files and add a table with aggregated results within the section.

Row 200: why are mail correspondences included?

Row 208: The period of published years is stated correctly. However, it is a bit misleading, as the search strategy stopped in mid 2019, so this year could not be entirely included.

Row 226: Often does not match with only one source mentioned. Moreover, the result does not match up with the results table, as there is only the cell patient satisfaction filled out and not cells for PROMs or PREMs.

Selective contracting: It would have been beneficial to start of with the terminological definitions/differences so that the reader understands the qualitative aspects of it and can follow the section easier.

Row 273-274: What is the source for it?

Row 279: What are the referred articles for PREMs here?

Prediction model: What kind of patient reported data is used here from the structure-process-outcome approach?

Quality of care: Analyses of QoC at large is lacking

Value chain framework:

The idea of using the value chain framework for the interpretation of the results sounds good. However, it is only mentioned three times in the article mainly by saying that the indicators are affecting all of the areas and without an in-depth interpretation of effects. It is entirely missing within the discussion part. The stated aim in the methods of using the framework to map patient reported data in the light of health insurer activity is therefore not fulfilled. It should be also reflected in the data charting, as this was described as one of the main points. If the framework should be incorporated into the article, a more in depth analysis should be performed. Otherwise it should be removed.

Limitations:

Some rudimental limitations of the work are missing, e.g. language restrictions that are leading towards a selection bias of included material, general bias within unsystematic approaches, here by choosing relevant organizations, influencing factors of different health systems, as they occur not only between high and middle/low income countries.

Row 425-427: The argumentation here is not convincing. A scoping review only makes sense, if an area is not exhaustively researched. Limiting then the search strategy towards previous work pieces seems a bit odd, as the scoping review aims to get more and broader insights into the topic before. A more grounded argumentation would have been beneficial.

Wording:

Row 275-277: Doesn’t some terminologies stated within the text more reflect applications like hospital ranking rather than terminologies?

Cost-effective: The article is using the term cost-effective inappropriately. In row 64 the authors stated that patient-provider collaboration are “cost-effective from a clinical perspective” and refer to a source from 2006. First of all it is questionable, if a statement of being cost-effective from 2006 is still applicable. Secondly, to what does a clinical perspective refer in this case in regards to cost-effectiveness? Thirdly, is it reliable to generalize the statement that patient-provider collaborations are cost-effective from an economics point of view (as the term is based on health economics)? Often, QoC or VBHC are not “good” in terms of the cost-effective ratio, but there are other ethical considerations that influence the decision about integration of interventions (or here patient-provider collaborations). Row 287-289 is here a little bit misleading, as it is meant that using patient reported data instead of “normally” gathered data for risk assessment could be cost-effective. In the article the ratio behind is not becoming clear. I would reframe it here.

6. PLOS authors have the option to publish the peer review history of their article (what does this mean?). If published, this will include your full peer review and any attached files.

Reviewer #1: No

Reviewer #2: **Yes: **Evalill Nilsson

Reviewer #3: No

---

## [Author Response · Author response to Decision Letter 0]

22 Oct 2020

Editor (E)

The main issue is that the frameworks used should be described better and, especially, applied more thoroughly throughout the manuscript. Also address inconsistencies in the scoping review methodologies used. Furthermore, focus on a critical discussion of the VBHC concept, address differences among health insurance systems in high-income countries, provide definitions or at least descriptions according to the reviewers’ comments (e.g. on clinical quality, selective contracting), consider revising the title (I am also not a native English speaker but it feels a bit odd, should “what” perhaps be “which”?) Please do not describe or discuss new literature/evidence from other studies in the results section but either in the methods or discussion (or introduction if suitable). Also consider to include only a selection of the main table in the main manuscript and include the remainder/full table in an appendix. Finally I am not entirely sure on the aspects included in the results section for “how”, as e.g. selective contracting is more of a purpose instead of exactly describing how patient-reported data are used, and ensure that the order of the different topics of what, how and why are always the same throughout the manuscript (e.g. in the title, how they are mentioned in the introduction, subsections in results and discussion).

We thank the Editor for highlighting the key aspects that needed revision to strengthen the quality and reading experience of our manuscript. One of the main revisions relates to how we report the summary of the documents retrieved in our searches, where we synthesized the information in a frequency table. While working on the results, we identified one document (a pamphlet retrieved via unstructured search) that did not meet the inclusion criteria, and thus, should have not been included. We updated the results accordingly. In the following pages we address the concerns highlighted by the Editor, which were also those of the Reviewers.

 

Reviewer #1 (R1)

The authors conducted a scoping review in order to map the evidence how insurers use patient-reported data. In doing so, the investigators highlighted patient-centeredness with respect to quality of care and value-based healthcare. The investigators specifically sought to understand why and how these data are collected and utilized. The review was conducted using the PRISMA extension for scoping reviews, and the analysis was framed using a value chain concept. The scoping review was conducted in a rigorous and transparent manner. The manuscript had a logical flow and the investigators explained, and provided citations for, the basis/framework for each step of the process. The manuscript was well-written, too. As a result, I do not perceive any areas of weakness with regard to the design or implementation of the manuscript. Although this may be outside of the scope of the manuscript, I was most interested in the potential “next steps” of this work based on the manuscript’s limitations. One area that might merit further thought is with regard to how stakeholders could implement a more consistent terminology of constructs. Furthermore, did the investigators think about broad differences in the way that citizens residing in different countries might have conceptualized the role of patient-reported measures and decision-making based on their nation’s health system and prevailing ideology regarding health? Finally, what role within the health system and in society should a health insurer play and is that role different in different nations? While these are questions have no easy answer, the investigative team is well-equipped to provide their opinions.

We thank the Reviewer for these very thoughtful insights regarding future research. We do agree that there are many unanswered questions in this field; some of which we mention in the Discussion section. One of our goals with conducting this scoping review was that of grasping a better understanding on the interactions between health insurers and insured people, and the role of patient-reported data in those interactions. Taking into consideration the findings of this study, we are conducting a new study which focuses on the societal role of a health insurer in a health system and how that shapes the extent to which insured people are involved in decision-making.

 

Reviewer #2 (R2)

(R2C1) The manuscript entitled “Why, what and how do health insurers use patient-reported data? Results of a scoping review” is well written and highlights an important topic of current interest, so far not studied to any extent. Given the breadth and variety of the topic, a scoping reviews seems like the best choice of method. The review process is well described and seems to be conducted in a sound way.

The title askes “why, what and how”. I understand why and how, and I guess “what do health insurers use patient-reported data” means which patient-reported data is being used, though I am not proficient enough in English to know if that is a grammatically correct expression? If “what” refers to something else, maybe it could be further explained?

We changed the title to “Understanding the use of patient-reported data by health care insurers: A scoping review” to accommodate the Reviewer’s comment.

(R2C2) Though QoC is by now a concept widely used globally, and the use of indicators of structure, process and outcome commonplace in evaluations, VBHC is more of a management trend (in some countries more than others), following earlier trends like TQM and lean production, and as such also subjected to criticism not acknowledged in the manuscript.

We now acknowledge in the text in a clearer manner that value-based healthcare may have different perspectives based on the taxonomy used, and thus, clarifying our use of such terminology. We believe that further discussion about the criticisms assigned to each taxonomy falls out of the scope of this work; rather, we use referencing that may be of interest to those who wish to seek further information on the topic. The text now reads as follows:

“Nowadays, it is commonplace to associate VBHC with care quality. Although a key component of quality, it is not necessarily the mainstream culture for measuring thereof. The VBHC agenda, similarly to the QoC, puts forward patients’ values regarding health and care outcomes, stressing their involvement in decision-making processes [4]. The construct of patient-centeredness emerges as a sub-dimension of those two concepts (QoC and VBHC) [5]. However, the inclusion of a people-centered perspective in VBHC is not without tensions as VBHC is a concept derived from management theories, with a clear conceptual focus on costs [6]. Hence, there can be a tension between the business model of a health care insurer oriented to optimizing the value for individual patients/insured versus optimizing the health of a population such as the group of individuals that pay their premium for the insurance. To strengthen people-centeredness and strive towards QoC and VBHC, health system stakeholders (e.g. health care insurers and care providers) should commit to the value agenda supported by intelligence on the healthcare system users’ needs, expectations, and preferences [7-9]. Hence, patient-reported data have become crucial to gain insight on one’s voice and inform the decisions of those key stakeholders.”

(R2C3) The decision to exclude the studies where the setting was not a high-income country, based on the assumption that a health insurance system in developing countries might differ greatly from that of developed countries, seems fair. However, also among high-income countries health insurance systems and their involvement in healthcare systems differ greatly, not acknowledged by the authors. Though I guess out of the scope of the present study it would be interesting to study the difference in the use of patient-reported outcomes divided by differences in health insurance systems and their involvement in healthcare systems.

The Reviewer addresses a valid viewpoint by highlighting the influence of context to understand differences on the use of patient-reported data among health insurers in high-income countries; we made this limitation clearer. The text now reads as follows:

“Also, limiting our search to high-income countries may have also introduced general bias. On the one hand, we did not consider studies from low-middle income countries because the use of patient-reported data in such contexts is yet limited [85]; on the other hand, we acknowledge that even in high-income countries, the extent to which patient-reported data are used may vary greatly, considering the role and involvement of a health insurer in the health system. Finally, given the study design, generalizability of results is limited; for example, contextual factors (e.g. the organization and digitalization of the healthcare system) vary greatly, and the extent to which these affect the uses and applications of patient-reported data are unknown.”

(R2C4) In the section “Data charting and analysis” it is described that indicators are organized in a traditional structure-process-outcome approach. For those not familiar with QoC and Donabedian, this might need an explanation, or at least a reference, maybe [1]?

We followed the Reviewer’s suggestion and added further explanation and two references. The text now reads as follows:

“We organized the list of indicators following the Donabedian’s healthcare quality model (structure-process-outcomes) [1, 30] given how widespread and familiar this model is across health systems and its stakeholders. This option could also facilitate a first approach to organize scattered information about the purposes and uses of patient-reported data by health insurers. Data mapped under structure highlight measures regarding the context and setting wherein care is delivered, and data under process highlight the interactions between a person and providers throughout the care trajectory; regarding data under outcomes, we organized information as clinical measures (referring to the diagnosis, treatment, and monitoring of a person), and patient-reported measures (PROMs, PREMs, and satisfaction measures). The focus of our work is on patient-reported data, but by using structure, process, and clinical measures as ancillary indicators in our work, we expected to have a better understanding on how patient-reported measures are used (as standalone or combined with other measures).”

(R2C5) In Table 1, I do not seem to find an explanation for the star* in PREM*? Furthermore, the definition of clinical quality is not stated. I am well aware of the problems with categorizing in this field, and reading Table 1 makes we wonder if all patient-reported data in the column PROM is actually reported as outcomes? I suspect some to be reported more as “patient characteristics”. “Health status” is a very generic term. I guess that the terms included in the table are the actual terms used in the papers (could be described), but if it is the authors interpretations, they should be defined/explained in more detail. Moreover, I wonder if only PROMS, PREMS and patient satisfaction are patient-reported? I guess that though the title implies that only patient-reported data are included in the study, the data regarding structure and process are mainly not patient-reported, but rather supportive data for your study. However, this is not very clearly stated.

To improve the reading experience, we removed the previous Table 1 (available now as Supplementary file). In this new Supplementary file, we clarify in a footnote that the terms used in the Table are the same as in the source document. We also clarify in-text the use of structure, process, and clinical indicators in support of achieving a better understanding about the use of patient-reported data by health insurers. That passage now reads as follows:

“Data mapped under structure highlight measures regarding the context and setting wherein care is delivered, and data under process highlight the interactions between a person and providers throughout the care trajectory; regarding data under outcomes, we organized information as clinical measures (referring to the diagnosis, treatment, and monitoring of a person), and patient-reported measures (PROMs, PREMs, and satisfaction measures). The focus of our work is on patient-reported data, but by using structure, process, and clinical measures as ancillary indicators in our work, we expected to have a better understanding on how patient-reported measures are used (as standalone or combined with other measures).”

(R2C6) In row 1, it is stated “RAND-36 survey, which included information on…”. I suspect that is not true, RAND 36 is the original, free version of SF-36, and does not include self-reported morbidity etc.

Thank you for flagging this issue; we revised the text.

(R2C7) The concept of selective contracting needs a short description, I think many readers are not familiar with the concept.

In hopes of clarifying the concept of selective contracting, we revised the text as follows:

“Selective contracting was discussed in five documents (12%) [60, 67-69, 74]. In general, selective contracting refers to the contractual agreement between a health insurer and a provider, where the former selects those providers that meet certain QoC expectations. The inclusion of QoC indicators is highly dependent on the availability of data; hence, the most common data used in these contracts are based on volume and costs [66, 69], and only recently some incorporate PROMs (and in a lesser extent, PREMs) [68, 74]. The use of structure, process, and outcome indicators for the purpose of selective contracting was discussed in Moes et al. (2019).”

(R2C8) I find your Discussion and Conclusion very apprehensive. The recommendation “In addition, health insurers should have a clear scope of the use of patient-reported measures” made me reflect on the fact that for so many years healthcare professionals have failed to accomplish just that. Could health insures achieve what health professionals could not, or will it be even more difficult for health insurers…

This is a fair reflection that we agree with. We included it in the text as follows:

“In addition, health insurers should have a clear scope about the use and actionability of patient-reported measures, and further involve insurees to the extent where it is feasible and deemed necessary. For many years now, there is a generalized consensus among healthcare providers and professionals for a greater involvement and engagement of people in decision-making towards a more people-centered health system. Albeit significant advances, we still fall short on that cornerstone. The extent to which lessons learned by health systems could be used and known obstacles could be overcome by health insurers remain overlooked and deserve further research.”

 

Reviewer #3 (R3)

(R3C1) In general, the topic of the article is well picked and scientifically relevant. The use of a scoping review is suitable for the aim of the article. Research question was addressed in methods and result section. However, there are some inconsistency in methods and results, so that question regarding external validity and pictured landscape of health insurer's use of patient reported data arises. In my opinion the concerns could be solved by clarification/specification of the explanations given. The main critic relies on the 13 pages table within the main text. This should be solved and adjusted due to the understanding and focus of the article. Also the frameworks used (structure-process-outcome as well as value chain) should not only be described better, but also be more applied within the text.

Following the comments of the Reviewer and those of the other Reviewers, we made changes throughout the text to address these issues, such as i) adding a revised version of Table 1 to synthesize our findings in a much simpler manner in hopes of improving the reading experience, rather than having an exhausting list of retrieved references (now available as supplementary material); and ii) excluding the use of the value chain framework to present and discuss the findings. We have also added an explanation to the use of the structure-process-outcome framework.

(R3C2) Scoping review methodology: The article stated that it was following Arksey and O’Malley (2005) stepwise framework. In the next section, they referred to the framework of Levac et al (2010) and later on to Silverman (2009), which seems a little bit inconsistent. Levac et al (2010) recommended and suggested for instance a update of the Arksey and O’Malley framework. The question arises, why different methodology were used or not only Levac was the entire foundation of the methodology. Probably it should be enough to frame this differently saying that Arksey was the basis, but recommendation of other authors was taken into account as well.

We thank the Reviewer’s suggestion to clarify this aspect of the methodology. We made the following in-text change:

“To enhance the quality of the methodology used, we based the review on the stepwise methodological framework suggested by Arksey and O'Malley (2005), while also taking into account the recommendations of other authors [25, 26].”

(R3C3) Structure-process-outcome approach: there was no explanation included within the methods section about the approach (even though it was claimed as ‘traditional’ and as the indicators for the research question). There was also no explanation of the rationale behind, the application in data extraction (e.g. why are outcomes differentiated into four subtypes and others not and are the suitable) and the suitability for the research question itself. However, it was referred to it in the result section entirely (good!), but, therefore, an introduction/explanation before is needed.

We included an earlier mention in-text to clarify this choice. It reads as follows:

“We organized the list of indicators following the Donabedian’s healthcare quality model (structure-process-outcomes) [1, 31] given how widespread and familiar this model is across health systems and its stakeholders. This option could also facilitate a first approach to organize scattered information about the purposes and uses of patient-reported data by health insurers. Data mapped under structure highlight measures regarding the context and setting wherein care is delivered, and data under process highlight the interactions between a person and providers throughout the care trajectory; regarding data under outcomes, we organized information as clinical measures (referring to the diagnosis, treatment, and monitoring of a person), and patient-reported measures (PROMs, PREMs, and satisfaction measures). The focus of our work is on patient-reported data, but by using structure, process, and clinical measures as ancillary indicators in our work, we expected to have a better understanding on how patient-reported measures are used (as standalone or combined with other measures).”

(R3C4) The search strategy for NIHR Journals Library, PDQ-Evidence, EBSCO/Health Business Elite and Cochrane Library is not attached within the supplement file. This should be added due to the PLOS One requirements. Moreover, NICE is stated within the supplement file, but not listed in the text for the systematic search.

Thank you for flagging these inconsistencies. We updated the search strategy in the Supplementary file.

(R3C5) Row 132-135: Rationale for expert interviews with Hungarian health insurance is unclear. Is this adoptable to other settings? Why was then the Hungarian language not included? Is asking only one insurance company not quite biased?

We decided to delete this information as it had no impact on our methodology nor the results and its interpretability. After the initial literature scan on terminology used by health insurers and early discussions among the research team to define the search strategy, we reached out to senior employees of an international insurance company in Budapest (Hungary) with whom we had discussed an early plan of this study. We asked these experts to weigh in with their working knowledge of the business on the most frequent or common terms used by health insurers to refer to patient-reported data, taking into consideration not only the company to which they worked for, but also their competitors.

(R3C6) Time limit for JSTOR: why was a time limit for JSTOR applied, but not for the other databases? Argumentation is flawed here.

JSTOR provides access to millions of academic journal articles spanning across 75 disciplines. Initial searches with no time limit retrieved many irrelevant hits off-topic and that could not be organized. When we considered more recent years by including a time limit (year 2000 onwards), the information retrieved were more on-topic and somewhat easier to work with. We clarified this in-text as follows:

“The search was not time bounded except for the JSTOR database. Not limiting the timeframe was producing a large number of hits off-topic to this review, which revealed to be unmanageable; hence, we limited the search from the year 2000 onward, where documents of potential relevance to the screening process started to emerge.”

(R3C7) Focus on PROMs and PREMs: in the introduction it is stated that a specific focus is on PROMs and PREMS. This was reflected by using the terms specifically in the search strategy for Pubmed and Embase. However, the focus was mainly seen in the introduction, later on the results showed a wider perspective. I would recommend to reframe it. Otherwise questions about possible bias, rational and differentiation towards other methods arises.

Thank you for flagging this inconsistency. We revised the aims of the study to clarify that we assumed a wider perspective when referring to patient-reported data.

(R3C8) Search terms: Search terms were mainly limited to Title/Abstract. The authors argued in the limitation section that Mesh terms/indexed words were not available for their specific topic. The question arises, why then the limitation on Title/Abstract was applied to mitigate the obstacles (especially because it is a scoping review). Furthermore, the limitation of Mesh Terms is in general obvious due to the time gap between new topics and the indexing process, so research in general should not entirely rely on them.

The key reason to limit the search to terms in the title/abstract is because broadening the search strategy resulted in a plethora of references off-topic. We sought to clarify this in-text, which now reads as follows: “We limited the search to keywords found in title and abstract to minimize the number of off-topic hits, which otherwise would have been unmanageable. The search strategy was adapted to each database and can be found as supplemental information (S2 File).”

(R3C9) The presented table is covering 13 pages in total in the main text. In my opinion, this is way too long for a scientific journal article and not benefitting the understanding of the article. I would highly recommend to move the table towards the supplemented files and add a table with aggregated results within the section.

We proceeded as suggested by the Reviewer.

(R3C10) Row 200: why are mail correspondences included?

These correspondences refer to email communication supporting references retrieved via the non-systematic search (e.g. searches on the webpages of health insurers). Often the information available on a health insurer website was insufficient and we felt the need to contact them in hopes of getting access to better information or other documents that could complement the initial webpage search. We tried to clarify this in-text; it now reads as follows: 

“For the non-systematic search, we included relevant grey literature such as webpages of insurers and third-party reports (S3 File). The search was performed between May 8th and June 18th, 2019. The documents retrieved were identified through online searches, reference mining, and recommendations from experts. The latter refers to contacts we have established via email (e.g. health insurers, insurance associations/federations, consultancy firms, or patient advocates) to direct us towards potential documents to complement the internet search. In total, 23 emails were sent to institutions that we believed could bring clarity or provide further information on top of what we had read on their webpages; 9 answers were received until the 18th of August 2019. A reminder was sent to all unanswered emails; we have received no reply to 14 emails and closed further contacts by the end of August 2019.”

(R3C11) Row 208: The period of published years is stated correctly. However, it is a bit misleading, as the search strategy stopped in mid 2019, so this year could not be entirely included.

We clarified the timeframe of the search strategy. The text now reads as follows: “The documents included in the study covered the period from 1996 to mid-2019 (Table 1)”.

(R3C12) Row 226: Often does not match with only one source mentioned. Moreover, the result does not match up with the results table, as there is only the cell patient satisfaction filled out and not cells for PROMs or PREMs. Selective contracting: It would have been beneficial to start of with the terminological definitions/differences so that the reader understands the qualitative aspects of it and can follow the section easier.

We cross checked the references to this issue flagged by the Reviewer. Also, we added the following in the text to clarify the term ‘selective contracting’: 

“Selective contracting was discussed in five documents (12%) [60, 67-69, 74]. In general, selective contracting refers to the contractual agreement between a health insurer and a provider, where the former selects those providers that meet certain QoC expectations. The inclusion of QoC indicators is highly dependent on the availability of data; hence, the most common data used in these contracts are based on volume and costs [66, 69], and only recently some incorporate PROMs (and in a lesser extent, PREMs) [68, 74]. The use of structure, process, and outcome indicators for the purpose of selective contracting was discussed in Moes et al. (2019).”

(R3C13) Row 273-274: What is the source for it?

This was an attempt of linking our results to the value chain framework. Given that we decided not to use the value chain, we removed that sentence.

(R3C14) Row 279: What are the referred articles for PREMs here? Prediction model: What kind of patient reported data is used here from the structure-process-outcome approach?

Thank you for flagging these missing references. We have now included them.

(R3C15) Value chain framework: The idea of using the value chain framework for the interpretation of the results sounds good. However, it is only mentioned three times in the article mainly by saying that the indicators are affecting all of the areas and without an in-depth interpretation of effects. It is entirely missing within the discussion part. The stated aim in the methods of using the framework to map patient reported data in the light of health insurer activity is therefore not fulfilled. It should be also reflected in the data charting, as this was described as one of the main points. If the framework should be incorporated into the article, a more in depth analysis should be performed. Otherwise it should be removed.

Following the suggestion of the Reviewer, we decided not to anchor our results and discussion in the value chain framework; hence, we removed it.

(R3C16) Limitations: Some rudimental limitations of the work are missing, e.g. language restrictions that are leading towards a selection bias of included material, general bias within unsystematic approaches, here by choosing relevant organizations, influencing factors of different health systems, as they occur not only between high and middle/low income countries.

We revised the text as follows:

“The findings of our review should be interpreted in light of some limitations. The heterogeneity of terminology, the use of an unsystematic search component, and language restriction may have introduced bias. To mitigate possible effects, the search strategy was informed by (but not limited to) the terminology used in other systematic reviews related to patient-reported data; we also assessed the extent to which documents retrieved via unsystematic search were aligned with those retrieved via systematic search. Also, limiting our search to high-income countries may have also introduced general bias. On the one hand, we did not consider studies from low-middle income countries because the use of patient-reported data in such contexts is yet limited [85]; on the other hand, we acknowledge that even in high-income countries, the extent to which patient-reported data are used may vary greatly, considering the role and involvement of a health insurer in the health system. Finally, given the study design, generalizability of results is limited; for example, contextual factors (e.g. the organization and digitalization of the healthcare system) vary greatly, and the extent to which these affect the uses and applications of patient-reported data are unknown.”

(R3C17) Row 425-427: The argumentation here is not convincing. A scoping review only makes sense, if an area is not exhaustively researched. Limiting then the search strategy towards previous work pieces seems a bit odd, as the scoping review aims to get more and broader insights into the topic before. A more grounded argumentation would have been beneficial.

We did not limit the search to previous works; rather, we were informed by other works on keywords that could be of beneficial use in our searches, taking into considerations our aims. We clarified this in the text that now reads as follows:

“The findings of our review should be interpreted in light of some limitations. The heterogeneity of terminology, the use of an unsystematic search component, and language restriction may have introduced bias. To mitigate possible effects, the search strategy was informed by (but not limited to) the terminology used in other systematic reviews related to patient-reported data; we also assessed the extent to which documents retrieved via unsystematic search were aligned with those retrieved via systematic search.”

(R3C18) Wording: Row 275-277: Doesn’t some terminologies stated within the text more reflect applications like hospital ranking rather than terminologies? 

We do agree that some expressions reflect more an application of quality reporting, rather than a terminology. However, as with other terminologies in this field, the cited expressions are often used interchangeably or as proxy to quality reporting.

(R3C19) Cost-effective: The article is using the term cost-effective inappropriately. In row 64 the authors stated that patient-provider collaboration are “cost-effective from a clinical perspective” and refer to a source from 2006. First of all it is questionable, if a statement of being cost-effective from 2006 is still applicable. Secondly, to what does a clinical perspective refer in this case in regards to cost-effectiveness? Thirdly, is it reliable to generalize the statement that patient-provider collaborations are cost-effective from an economics point of view (as the term is based on health economics)? Often, QoC or VBHC are not “good” in terms of the cost-effective ratio, but there are other ethical considerations that influence the decision about integration of interventions (or here patient-provider collaborations). Row 287-289 is here a little bit misleading, as it is meant that using patient reported data instead of “normally” gathered data for risk assessment could be cost-effective. In the article the ratio behind is not becoming clear. I would reframe it here.

We have removed this reference to the term cost-effective.

---

## [Decision Letter · Decision Letter 1]

14 Dec 2020

Understanding the use of patient-reported data by health care insurers: A scoping review

PONE-D-20-09776R1

Dear Dr. Brito Fernandes,

We’re pleased to inform you that your manuscript has been judged scientifically suitable for publication and will be formally accepted for publication once it meets all outstanding technical requirements.

Kind regards,

Mathieu F. Janssen, Ph.D.

Academic Editor

PLOS ONE

Additional Editor Comments (optional):

Feel free to take some final minor suggestions on board provided by reviewer 3.

Reviewers' comments:

Reviewer's Responses to Questions

**Comments to the Author**

1. If the authors have adequately addressed your comments raised in a previous round of review and you feel that this manuscript is now acceptable for publication, you may indicate that here to bypass the “Comments to the Author” section, enter your conflict of interest statement in the “Confidential to Editor” section, and submit your "Accept" recommendation.

Reviewer #1: All comments have been addressed

Reviewer #3: All comments have been addressed

2. Is the manuscript technically sound, and do the data support the conclusions?

Reviewer #1: Yes

Reviewer #3: Yes

3. Has the statistical analysis been performed appropriately and rigorously? 

Reviewer #1: N/A

Reviewer #3: N/A

4. Have the authors made all data underlying the findings in their manuscript fully available?

Reviewer #1: Yes

Reviewer #3: Yes

5. Is the manuscript presented in an intelligible fashion and written in standard English?

Reviewer #1: Yes

Reviewer #3: Yes

6. Review Comments to the Author

Reviewer #1: The authors have responded to the comments of the reviewers in a comprehensive and thoughtful manner. I do not have any additional comments or feedback.

Reviewer #3: The authors revised and incorporated the suggestions of the reviewers in their manuscript sufficiently. Now, the article holds for internal validity. The adjusted research questions and method section are consistent and precisely tailored towards the results presented. The discussion analyses the results thoughtfully and transferred the findings onto a broader scale. The limitation of the study are transparently represented.

There are three fine points I would like to further comment on:

Value-based healthcare: The clarification of the concept within the introduction section is reasonable. However, as introduced by Porter (original source is not mentioned within the paper), costs are a driver of value within his introduced formula of outcome/costs=value. This indicates, that a greater outcome of a more expensive intervention could produce a greater value than a less expensive intervention with a great outcome loss under certain circumstances. Therefore, outcome and costs are dependent resulting in theory towards efficient use of resources, so that the mentioned conceptual focus on costs alone as a description of the concept is, from my point of view, too simplified. Furthermore, as described by the authors, VBHC is an emerging trend within health care systems, so that a more exhaustive discussion of (missing) VBHC in the context of patient-reported data in the literature would have been interesting

Search strategy: The argumentation of the time limit for JSTOR is pragmatic and reasonable, but a source about the emerging number of papers from 2000 onwards would have increased the convincement of the sentence.

Lastly, as a formal point, the text formatting is not consistent throughout the manuscript, but I guess this is solved by the publisher itself.

Overall, as the points raised are more part of a scientific discussion as well as editorial remarks rather than serious weaknesses of the article, I would recommend the acceptance of the article (optionally with some small, final adjustments).

7. PLOS authors have the option to publish the peer review history of their article (what does this mean?). If published, this will include your full peer review and any attached files.

Reviewer #1: No

Reviewer #3: No

---

## [Editor Report · Acceptance letter]

16 Dec 2020

PONE-D-20-09776R1 

Understanding the use of patient-reported data by health care insurers: A scoping review 

Dear Dr. Brito Fernandes:

I'm pleased to inform you that your manuscript has been deemed suitable for publication in PLOS ONE. Congratulations! Your manuscript is now with our production department. 

Kind regards, 

on behalf of

Dr. Mathieu F. Janssen 

Academic Editor

PLOS ONE